# Unsupervised Exercise Intervention vs. Adherence to a Mediterranean Diet Alone: The Role of Bioelectrical Impedance Vector Analysis and Cardiovascular Performance in Liver-Transplanted Recipients

**DOI:** 10.3390/nu16020190

**Published:** 2024-01-05

**Authors:** Gabriele Mascherini, Marco Corsi, Edoardo Falconi, Álex Cebrián-Ponce, Pietro Checcucci, Antonio Pinazzi, Domenico Russo, Stefano Gitto, Francesco Sofi, Laura Stefani

**Affiliations:** 1Department of Experimental and Clinical Medicine, University of Florence, 50134 Florence, Italy; gabriele.mascherini@unifi.it (G.M.); marco.corsi@unifi.it (M.C.); edoardo.falconi@unifi.it (E.F.); pietro.checcucci@unifi.it (P.C.); dr.antoniopinazzi@gmail.com (A.P.); domenico.russo@unifi.it (D.R.); stefano.gitto@unifi.it (S.G.); francesco.sofi@unifi.it (F.S.); 2Barcelona Sports Sciences Research Group, Institut Nacional d’Educació Física de Catalunya (INEFC), University of Barcelona (UB), 08038 Barcelona, Spain; acebrian@gencat.cat

**Keywords:** exercise prescription, nutrition, body composition, BIVA, cardiopulmonary exercise testing, CPET, solid organ transplant

## Abstract

(1) Background: Cardiovascular disease is one of the leading causes of mortality after liver transplantation. Body composition and cardiovascular performance assessment represent a potential approach for modulating lifestyle correction and proper follow-up in chronic disease patients. This study aimed to verify the additional role of an unsupervised physical activity program in a sample of male liver transplant recipients who follow the Mediterranean diet. (2) Methods: Thirty-three male liver transplant recipients were enrolled. Sixteen subjects followed a moderate-intensity home exercise program in addition to nutritional support, and seventeen received advice on the Mediterranean diet. After six months, bioelectrical vector impedance analysis (BIVA) and cardiopulmonary exercise testing (CPET) were performed. (3) Results: No differences in CPET (VO_2_ peak: exercise 21.4 ± 4.1 vs. diet 23.5 ± 6.5 mL/kg/min; *p* = 0.283) and BIVA (Z/H: exercise 288.3 ± 33.9 vs. diet 310.5 ± 34.2 Ω/m; *p* = 0.071) were found. Furthermore, the BIVA values of resistance correlate with the submaximal performance of the Ve/VCO_2_ slope (R = 0.509; *p* < 0.05) and phase angle with the maximal effort of the VO_2_ peak (R = 0.557; *p* < 0.05). (4) Conclusions: Unsupervised physical exercise alone for six months does not substantially modify liver transplant recipients’ cardiovascular performance and hydration status, despite their adherence to a Mediterranean diet. The body composition analysis is useful to stratify the risk profile, and it is potentially associated with better outcomes in transplanted subjects.

## 1. Introduction

Solid organ transplantation is a therapeutic strategy in the end-stage of the disease or in emergency cases. Surgery and care techniques have improved in recent years, ensuring increasingly greater survival after transplantation. In the last ten years in Italy, there has been a 26% increase in the number of solid organ transplants, among which liver transplants have had one of the largest increases, standing at 44% [1].

Lifestyle, combining regular physical activity and adherence to a healthy diet, represents a treatment for many chronic diseases [2]. Cardiovascular disease represents a major cause of mortality after liver transplantation [3], and it is long-established that healthy habits reduce risk factors by improving physical fitness parameters related to health in solid organ transplant recipients [4].

Physical inactivity is a crucial driver of progression and adverse outcomes in liver diseases [5]. In particular, exercise’s role in reducing mortality is currently debated [6] in liver transplant recipients. After a successful liver transplantation, the patients show impaired exercise capacity and fatigue due to a minor effort [7]. This fatigability is explained by prolonged bed rest after transplantation, immunosuppressive drugs, associated comorbidities (e.g., obesity, diabetes, hyperlipidemia, hypertension, and metabolic syndrome), and sarcopenia [8]. In this context, the evidence suggests that exercise programs improve two parameters related to cardiovascular adverse events after liver transplantation: maximal oxygen uptake [9,10] and body composition [9]. Therefore, physical activity is increasingly recommended as a therapeutic approach after solid liver transplantation [11]. 

Cardiovascular risk factors after a liver transplant are also modifiable by eating habits. These patients consume a high-energy, low-quality diet in the long term [12], and they do not adhere to the dietary guidelines for cardiovascular disease prevention [13]. Studies initially focused on controlling caloric, fat, and protein intake, also using supplements [9,14]. More recently, attention has begun, with promising results, to focus on promoting the Mediterranean diet to increase healthy eating habits after liver transplantation [15,16].

In the three years following liver transplantation, there is an average weight gain of about 10 kg, and most patients are overweight or obese [17,18]. Furthermore, sarcopenia is present in more than half of liver transplant recipients [19]; therefore, one might wonder whether, in some cases, weight gain could be considered sarcopenic obesity [20]. An early and inappropriate increase in fat mass characterizes changes in the body composition of transplant recipients. In contrast, the restoration of cell mass and fluid distribution appears to occur more slowly and is incomplete [21]. Recently, patients on the liver transplant waiting list have been evaluated using bioelectrical impedance vector analysis (BIVA). The placement of subjects in the RXc graph quadrants in vector impedance interpretation has been found to have a prognostic factor [22].

Body composition and cardiovascular performance assessment represent an approach for modulating lifestyle correction and carrying out a proper follow-up. This study aimed to verify the invention’s effectiveness on lifestyle by comparing the promotion of a Mediterranean diet alone vs. a Mediterranean diet plus an unsupervised physical activity program in a sample of male liver transplant recipients. 

## 2. Materials and Methods

### 2.1. Participants

Thirty-three male liver transplant recipients, aged 61.4 ± 8.0 years and 1.2 ± 0.7 years post-transplant, were enrolled in this study from February 2021 to January 2023. Inclusion criteria were to be transplanted for at least one year and clinically stable (e.g., absence of liver-related complications in the previous six months, including acute rejection episodes and increased serum transaminases two times the upper limit). Exclusion criteria were combined transplantation, re-liver transplantation, physical limitations, cardiovascular contraindications to exercise, and psychiatric or severe debilitating neurological disorders. A total of 19 participants had mild or moderate hypertension and were under antihypertensive treatment (calcium channel blockers, ACE inhibitors, or ARBs); 14 participants had no hypertension. All participants assume immunosuppressive therapy, including drugs such as calcineurin inhibitors (Ciclosporin or Tacrolimus), in combination with Mycophenolate or Everolimus, and steroids (Methylprednisolone). Comorbidities, such as diabetes, hypertension, or other metabolic diseases, were not a reason for exclusion. None of them assumed beta-blockers. This study was conducted in accordance with the Declaration of Helsinki and subsequent modifications and approved by the ISRCTN registry (study ID: ISRCTN66295470, 19 January 2017). All participants provided written, informed consent.

### 2.2. Physical Exercise Program and Dietary Intervention

Sixteen transplanted subjects followed a tailored home-based exercise program with moderate intensity with nutritional support. An unsupervised physical exercise program was chosen to prevent participants from traveling to a specific health club to carry out the training to achieve a lower probability of absence from exercise and greater adherence to the program in the long run.

The physical exercise program consisted of mixed physical activity (endurance and resistance exercise) for 60 min thrice weekly, following the American College of Sports Medicine guidelines [23]. Endurance exercises were prescribed for up to 30 min with an intensity of around 60% of the maximal heart rate. In particular, the heart rate range indicated was established individually using the Karvonen formula [24]. Resistance exercise involved training eight major muscle groups for three sets of ten repetitions for the remaining 30 min after the endurance exercise. The exercises were chosen based on the possibility of being performed safely at home (such as a bodyweight squat and glute bridge for the lower limbs, a lateral raise, and a biceps curl for the upper limbs). Furthermore, a qualified kinesiologist demonstrated resistance exercise, followed by repetition by the patient as a learning test. In order to verify adherence to the prescribed physical exercise, the International Physical Activity Questionnaire (IPAQ) was used [25]. Data collected with the IPAQ were reported as a continuous measure and expressed as METs/min/week (MET: metabolic equivalent of task; one MET is defined as the amount of oxygen consumed at rest, which corresponds to 3.5 mL of O_2_ per kg of body weight × min). Achieving at least a value of 600 was considered moderately active, and 1500 was considered vigorously active. 

Despite being physically active, the remaining seventeen subjects were outside of a structured exercise program and received indications for the Mediterranean diet. Specifically, the recommendation was about long-term nutritional support after liver transplantation [26]. In particular, it was recommended to limit fat intake and consume adequate amounts of lean protein to promote muscle development, in parallel with ensuring an adequate calorie intake to avoid protein utilization as an energy source. Finally, no added salt (a daily maximum of 3 g of sodium) was recommended to prevent water retention. In order to verify adherence to nutritional advice on the Mediterranean diet, the MEDI-LITE score was used [27]. Subjects reporting a score higher than 8.5 on the MEDI-LITE questionnaire should be considered adherents to the Mediterranean diet.

Six months after receiving lifestyle recommendations, exercise prescriptions, or Mediterranean diet indications, the evaluations were performed for both groups to establish the effectiveness of each program. Body composition assessment and cardiopulmonary exercise testing (CPET) were performed to measure the nutritional and hydration status as well as the cardiovascular and respiratory performances.

### 2.3. Cardiopulmonary Exercise Testing

The cardiopulmonary exercise testing (CPET) was conducted by an electromagnetic brake cycle ergometer (Ergoline) and a specific gas measure machine (COSMED Quark). Every participant was invited to avoid strenuous physical exertion the day before the evaluation and to refrain from eating solid foods or carbohydrate drinks at least three hours before the test. The CPET was performed in the morning in a room with controlled conditions (temperature 18–24 °C; humidity 30–60%). The protocol was established based on sex, age, height, and weight, and the training evaluation was declared. The ramp was also individualized on predicted weight values to achieve muscle exhaustion between 8 and 12 min [28]. Participants were equipped with an orofacial mask connected to a gas measuring device [29]. Exhaled CO_2_ and O consumed were measured using the breath-by-breath method. The lowest possible increase in watts (1, 2, or 5) was set for each ramp to obtain the most linear possible increase in load and, consequently, a more physiological response. After the first 3 min of warming up, by cycling without load at a cadence of 50 rpm (revolutions per minute), the test started. At the beginning of the actual effort, cycling was required at a cadence between 60 and 80 rpm until muscle exhaustion was reached. The test ended when the participants could no longer maintain their cycling cadence despite verbal encouragement [30]. The test was considered maximal with at least two of the following criteria:Respiratory exchange ratio (RER) > 1.10.Heart rate (HR) > 85% according to age.Exercise duration between 8 and 12 min.

CPET was stopped in cases of cardiovascular signs and symptoms (complex ventricular arrhythmias, a drop in systolic blood pressure, dizziness, etc.). The 12-lead ECG and oxygen saturation were monitored continuously.

During the test, oxygen consumption (VO_2_), carbon dioxide production (VCO_2_), tidal volume (TV), respiratory rate (RR), minute ventilation (VE), heart rate (HR), and workload (WL, in watt) were obtained. The two ventilatory thresholds (VT1 and VT2) were indirectly determined using the combination of V-slope and the ventilatory equivalents approach. In addition, other variables analyzed were the relationship between oxygen consumption and heart rate (oxygen pulse, VO_2_/HR), minute ventilation/carbon dioxide production slope (VE/VCO_2_ slope), and the relationship between oxygen consumption and workload [31] (VO_2_/WL slope, as a measure of circulatory efficiency).

### 2.4. Body Composition Analysis

Bioimpedance analysis was chosen to evaluate body composition. The bioelectrical parameters of resistance (R) and reactance (Xc) were measured with a BIA 101 Anniversary Sport Edition analyzer (Akern Srl, Florence, Italy) emitting an alternating sinusoidal current of 400 mA at 50 kHz (±0.1%). Before each evaluation, this device was calibrated with a known impedance circuit provided by the manufacturer.

The assessments were carried out according to the guidelines, with arms and legs abducted to prevent contact with the body. The measurements were recorded after a 5 min stabilization period, in which the participants remained still to ensure a homogeneous distribution of body fluids. Injector electrodes were placed on the dorsal surface of the right hand (proximal to the third metacarpophalangeal joint) and right foot (proximal to the third metatarsophalangeal joint). The sensing electrodes were placed approximately 5 cm from the injector to prevent interaction between the electric fields and to avoid overestimating the impedance values.

Impedance (Z) was calculated as (R^2^ + Xc^2^)1/2, and phase angle (PhA) as tan^−1^ (Xc/R · 180°/π). R, Xc, and Z were adjusted by height (R/H, Xc/H, Z/H). According to classic BIVA, Z/H is inversely related to total body water (TBW) [32]. In contrast, vector direction indicates cellular health and cell membrane integrity and is inversely related to the extracellular/intracellular water (ECW/ICW) ratio [33]. All interpretations should be based on the interpretation of Z/H and PhA jointly, along with the vector position on the Resistance-Reactance (RXc) graphs [34]. In this graph, shifts in vectors parallel to the major ellipse axis indicate differences in tissue hydration (a longer vector indicates less fluid, while a shorter vector indicates more body fluids). Shifts in the vector parallel to the minor axis of the ellipses indicate differences in cell mass and ECW/ICW ratio (a shift to the left indicates an increase in cell mass and a reduction in the ECW/ICW ratio, while a shift to the right indicates a reduction of cell mass and an increase in the ECW/ICW ratio [35]).

### 2.5. Statistical Analysis

Descriptive analysis was calculated, and the data are presented as mean ± standard deviation (SD). After testing each variable for the normality of the distribution (Shapiro–Wilks test), differences in all the variables were tested using a student’s *t*-test in cases of normal distribution and a Mann–Whitney test in cases of the data not being distributed normally. The relative effect sizes (ES) were calculated using Cohen’s d [36] to estimate the relevance of the differences analyzed. According to Cohen, ES is defined as small (≤0.20), medium (≤0.50), and large (≤0.80). Pearson’s correlation test (Spearman’s test for not normally distributed values) was applied to examine the relationships between BIVA and CPET. RXc point graph was used to plot the points of the subjects regarding the 50%, 75%, and 95% tolerance ellipses of the healthy reference population [37]. RXc mean graph and two-sample Hotelling’s T^2^ test were used to check the differences in the complex vector between groups. The significance level was set at *p* < 0.05. SPSS (Chicago, IL, USA, ver. 21) and BIVA software [38] were used for data analysis. 

## 3. Results

Thirty-three males who had undergone liver transplantation 1.2 ± 0.7 years previously were enrolled. From this total sample, 16 subjects (age 61.7 ± 8.8 years, BMI 26.5 ± 3.3 kg/m^2^) performed the unsupervised exercise program, and 17 subjects (age 61.1 ± 7.5 years, BMI 26.9 ± 1.7 kg/m^2^) were adherent to nutritional advice relating to the Mediterranean diet with a score > 8.5.

The physical exercise group reported higher but not statistically significant values of weekly METs from IPAQ (1269.3 ± 1272.4 vs. 951.9 ± 875.5; *p* = 0.408, ES = 0.290), and the diet group reported higher but not statistically significant values of the MED-LITE score (11.7 ± 2.1 vs. 10.4 ± 2.3; *p*= 0.236, ES = 0.590).

The results relating to cardiorespiratory performance, assessed with the CPET, show no differences between the two groups (Table 1). 

Table 2 and Figure 1 compare body composition assessed with the impedance vector analysis. No significant differences between the two groups were found, even if the different positioning of the two groups on the RXc mean graph (Figure 1B) seems to be more attributable to the R component (physical exercise group = 286.6 ± 33.8 vs. diet group = 308.8 ± 33.9; *p* = 0.069, ES = 0.656) rather than the Xc component (exercise group = 30.1 ± 5.7 vs. diet group = 32.3 ± 6.4; *p* = 0.314, ES = 0.363).

Finally, the analysis of the correlations between the BIVA and CPET parameters shows how lower resistance values (R) are linked to better submaximal performance. At the same time, the PhA provides information on maximal performance with a direct relationship with oxygen consumption and ventilation at peak effort (Table 3). 

In detail, analyzing these correlations divided based on the two study groups, the R parameter appears to have greater relevance for the performance of the exercise group about VO_2_/HR (R = −0.648; *p* < 0.05), Ve/VCO_2_ slope (R = 0.573; *p* < 0.05), and peak VE (R = 0.516; *p* < 0.05). While in the diet group, the phase angle shows a greater relationship with power (R = 0.568; *p* < 0.05) and with the VE peak (R = 0.579; *p* < 0.05).

## 4. Discussion

This study investigates the potential effectiveness of the additional role of unsupervised exercise programs in a group of liver-transplanted males that follow nutritional advice. Under stable clinical conditions, the study sample was divided into two groups. One group was encouraged to lead healthy eating habits with dietary advice based on the Mediterranean diet, and the other group was encouraged to lead an active lifestyle in addition to nutritional advice. Both groups performed their program for six months, and then evaluations were done to establish the effectiveness of each program. 

Based on the questionnaires, the two groups should be considered moderately physically active (around 1000 METs with IPAQ) and follow the Mediterranean diet (both groups have a MED-LITE score above 8.5). The IPAQ questionnaire was recently administered to liver transplant candidates to assess how physical activity levels are related to physical performance and frailty [39]. In this study, patients showed low-to-medium levels. However, in another two studies [40,41], patients who have already undergone a liver transplant reported higher values than those in the present study. In particular, Kotarska et al. [40] showed rather high values compatible with daily workouts of vigorous intensity. The results obtained from our study show that the declared levels are, although not significantly, slightly higher in the exercise group and compatible with moderate-intensity activity. Therefore, the exercise prescription proposed in this study may be reasonable because adult transplant recipients are among the most sedentary of all populations with chronic disease, with a daily step count of 3164 ± 2842 steps [5]. The values MED-LITE recorded in this study align with the data recently published on a larger sample of the same geographical area following indications on the Mediterranean diet [16]. The group that received only indications on the Mediterranean diet, albeit not significantly, showed higher values than the group that also followed the exercise program. The exercise group also obtained a mean MED-LITE value >8.5 and, therefore, can still be considered adhering to the principles of the Mediterranean diet; however, the greater effort required by physical exercise could compromise, at least in part, the adherence to the nutritional advice. 

Interestingly, the two groups had no significant differences in oxygen consumption parameters and the other parameters of CPET. However, the VO_2_ peak value reported in the present or can be considered above the mean of 20.5 mLO_2_/kg/min reported at the end of three previous supervised exercise interventions performed in three randomized controlled trials [9,10,42]. In addition, Totti et al. [43] report that 12 months of supervised physical exercise with a training program comparable to that of the present study did not increase the VO_2_ peak in a group of liver transplant recipients. Therefore, the cardiorespiratory performances recorded in the present study allow for establishing this sample as a liver-transplant recipient with a high level of cardiovascular fitness. Furthermore, the current ACSM guidelines [23] do not indicate the physical activity regimen post-transplantation, and there is yet to be evidence of the stronger benefits of an intensive physical activity program in this particular population of patients. 

The present study reports the bioelectrical impedance vector analysis assessment in liver transplant recipients for the first time. The data suggest a medium mortality risk profile due to incorrect water distribution. A study [22] shows that patients on the liver transplant waiting list have vector impedance placed mainly in the lower quadrants of the RXc graph, indicating increased body water content. In addition, vector placement in the lower right quadrant and ascites or edema were independent risk factors for the wait list and 1-year post-transplant mortality. The placement in the RXc graph of the sample of this study (Figure 1) shows how the subjects are, on average, in the center of the reference ellipse (Figure 1B), with only four subjects in the lower right quadrant (Figure 1A) within the 50th percentile and, therefore, with a more favorable prognostic perspective. Another study, conducted by the same research center as the present study, performed bioelectrical impedance vector analysis on thirteen kidney transplant patients who performed an unsupervised exercise program [44]. Their placement is reported to be, on average, within the 50th percentile of the lower right quadrant. However, it must be considered that the state of hydration and the intra/extracellular distribution may differ due to the different transplanted solid organs, where renal function in the hydro-salt balance plays a leading role compared to liver function.

The relationship between bioelectrical parameters and physical performance in liver transplant recipients is a new direction of study never investigated before, especially in this population. Lower resistance values are compatible with a greater state of hydration, which could allow better cellular function and more efficient thermoregulation and, therefore, better submaximal endurance performance. At the same time, the role of PhA in muscle performance is well established in the literature, even in subjects with chronic disease [45]. The present study confirms that higher PhA values that reflect greater cellularity support physical performance close to maximum individual effort.

Lifestyle correction plays a relevant role as a therapeutic intervention in many non-communicable chronic diseases, and it has also recently been promoted in post-transplanted subjects [46]. Lifestyle intervention is normally proposed as diet correction and/or physical activity combination. However, more data should be available regarding the efficacy of the additional component of the unsupervised exercise activity in the presence of sufficient adherence to the Mediterranean diet.

This study has, therefore, some strengths and points of interest. Firstly, the sample size aligns with and even exceeds existing similar studies. Secondly, the same healthcare professionals trained in CPET and BIVA evaluated all subjects with the same instrumentation. Thirdly, the evaluation methodologies were direct measurements and not an indirect estimation: cardiorespiratory performance was measured through a CPET, unlike other studies, which estimated VO_2_ peak through the six-minute walking test, while body composition was evaluated with the BIVA method rather than an estimation through a regression equation derived from anthropometric parameters.

The authors are aware that the absence of an initial evaluation of the enrolled subjects does not allow a comparison with the values obtained after six months of joining the program. However, baseline assessments are not compatible with the care pathway. In addition, the values obtained from this study sample at CPET allow us to speculate that they are compatible with a healthy lifestyle.

## 5. Conclusions

In summary, unsupervised physical exercise appears feasible and sustainable; however, the long-term efficacy should be studied. The data are in any case in agreement to a frailty in post-transplanted subjects and to the importance of physical activity to improve quality of life and mitigate the CV risk. Despite the absence of specific guidelines or recommendations in this category, promoting more intensive and potentially supervised training is reasonable. Body composition analysis seems fundamental in the initial phase to stratify the risk profile due to excessive water compartmentalization. In addition, the results obtained in this study suggest that an approach based on a single aspect of the lifestyle, like diet and moderate physical activity, could not be sufficient to influence cardiovascular parameters.

## Figures and Tables

**Figure 1 nutrients-16-00190-f001:**
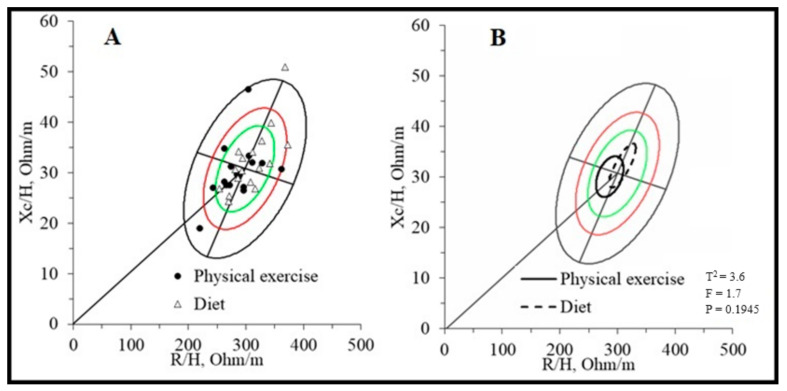
BIVA Resistance-Reactance (RXc) graphs of liver transplant recipients. Legend: (**A**). RXc point graph; (**B**). RXc mean graph. The green, red and black ellipses represent the 50%, 75% and 95% tolerance respectively.

**Table 1 nutrients-16-00190-t001:** Data obtained from liver transplant recipient samples at CPET.

	Physical Exercise (*n* = 16)	Diet (*n* = 17)	*p*-Value	ES
VO_2_ peak (mL/kg/min)	21.4 ± 4.1	23.5 ± 6.5	0.283	0.386
VO_2_/HR	12.5 ± 2.0	12.7 ± 2.8	0.790	0.082
Power (Watt)	128.8 ± 34.0	142.4 ± 52.4	0.385	0.308
Ve/VCO_2_ Slope	33.6 ± 6.0	33.6 ± 6.1	0.996	0.000
VO_2_/WL	10.2 ± 0.9	10.6 ± 1.3	0.295	0.358
HR VT1 (bpm)	97.8 ± 15.3	100.8 ± 15.8	0.574	0.192
HR VT2 (bpm)	120.9 ± 21.4	124.2 ± 17.8	0.627	0.167
HR max (bpm)	153.6 ± 18.0	152.9 ± 15.8	0.909	0.041
VE peak (L/min)	70.8 ± 14.5	80.0 ± 22.8	0.183	0.481
SBP rest (mmhg)	121.9 ± 10.3	123.8 ± 16.0	0.682	0.141
DBP rest (mmhg)	73.4 ± 11.4	74.7 ± 7.2	0.702	0.136
SBP peak (mmhg)	167.2 ± 21.6	165.3 ± 20.1	0.796	0.091
DBP peak (mmhg)	71.9 ± 15.2	72.4 ± 11.3	0.919	0.037

Legend: Data are expressed as mean ± s.d. VO_2_ peak = oxygen consumption achieved at peak performance; VO_2_/HR = ratio between oxygen consumption and heart rate; Ve/VCO_2_ slope = minute ventilation/carbon dioxide production slope; VO_2_/WL = ratio between oxygen consumption and workload; HR VT1 = heart rate corresponding to the first ventilatory threshold; HR VT2 = heart rate corresponding to the second ventilatory threshold; HR max = heart rate at the peak of exertion; VE peak = peak ventilation; SBP rest = systolic blood pressure at rest; DBP rest = diastolic blood pressure at rest; SBP peak = systolic blood pressure at the peak of exertion; DBP peak = diastolic blood pressure at the peak of exertion.

**Table 2 nutrients-16-00190-t002:** Bioimpedance vector analysis (BIVA) data of the two groups of patients undergoing liver transplantation.

	Physical Exercise (*n* = 16)	Diet (*n* = 17)	*p*-Value	ES
R/H (Ω/m)	286.6 ± 33.8	308.8 ± 33.9	0.069	0.656
Xc/H (Ω/m)	30.1 ± 5.7	32.3 ± 6.4	0.314	0.363
Z/H (Ω/m)	288.3 ± 33.9	310.5 ± 34.2	0.071	0.652
PhA (°)	6.0 ± 1.0	5.9 ± 0.7	0.847	0.116

Legend: Data are expressed as mean ± s.d. R/H = ratio between resistance and height; Xc/H = ratio between reactance and height; Z/H = ratio between impedance and height; PhA = phase angle.

**Table 3 nutrients-16-00190-t003:** Matrix of correlations between bioimpedance vector analysis (BIVA) and CPET performance parameters.

	R/H	Xc/H	Z/H	PhA
VO_2_ peak	−0.013	0.294	0.019	0.557 *
VO_2_/HR	−0.392 *	−0.107	−0.391 *	0.230
Power	−0.237	0.101	−0.233	0.509 *
Ve/VCO_2_ slope	0.509 *	0.112	0.506 *	−0.222
VO_2_/WL	0.403 *	0.267	0.403 *	0.132
HR VT	0.183	0.369 *	0.316	0.377 *
HR max	−0.002	0.053	0.001	0.213
VE peak	0.087	0.392 *	0.092	0.506 *
SBP rest	−0.109	−0.253	−0.114	−0.214
DBP rest	0.198	−0.075	0.193	−0.135
SBP peak	0.072	−0.119	0.072	−0.116
DBP peak	0.278	0.296	0275	0.177

Legend: R/H = ratio between resistance and height; Xc/H = ratio between reactance and height; Z/H = ratio between impedance and height; PhA = phase angle; VO_2_ peak = oxygen consumption achieved at peak performance; VO_2_/HR = ratio between oxygen consumption and heart rate; Ve/VCO_2_ slope = minute ventilation/carbon dioxide production slope; VO_2_/WL = ratio between oxygen consumption and workload; HR VT = heart rate corresponding to the ventilatory threshold; HR max = heart rate at the peak of exertion; VE peak = peak ventilation; SBP rest = systolic blood pressure at rest; DBP rest = diastolic blood pressure at rest; SBP peak = systolic blood pressure at the peak of exertion; DBP peak = diastolic blood pressure at the peak of exertion. * *p* < 0.05.

## Data Availability

Data can be obtained from Gabriele Mascherini on gabriele.mascherini@unifi.it. The data are not publicly available due to are part of an ongoing study.

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
