# Peer review of "Unsupervised Exercise Intervention vs. Adherence to a Mediterranean Diet Alone: The Role of Bioelectrical Impedance Vector Analysis and Cardiovascular Performance in Liver-Transplanted Recipients"

_nutrients, 2024, doi:10.3390/nu16020190_

Round 1

Reviewer 1 Report

Comments and Suggestions for Authors

1. It is necessary to explain why we approached liver transplantation. 2. The type of nutrients ingested is unknown. Regarding the exercise method, it is also necessary to explain why this method was chosen. 3. As a result, the relationship between exercise and diet is already known, and it is difficult to differentiate this from previous content. 4. I don't understand whether you are referring to the heart or the liver.

Comments on the Quality of English Language

I'm not a native speaker so I can't discuss

Author Response

Reviewer 1:

The authors would like to thank the reviewer for appreciating our work and suggestions provided to improve our manuscript.

Changes made to the manuscript are highlighted in red.

Below are the answers to the reviewer’s comments.

  1. It is necessary to explain why we approached liver transplantation.

Answer: Thanks for the comment. The introduction section was implemented with a description of the transplant activity in Italy with particular reference to liver transplants. The sentences added was between line 34-38:

“Solid organ transplantation is a therapeutic strategy in the end-stage of the disease or emergency cases. Surgery and care techniques have improved in recent years, ensuring increasingly greater survival after transplantation. In the last ten years in Italy, there has been a 26% increase in the number of solid organ transplants, among which liver trans-plants have had one of the largest increases, standing at 44% [1].”

  1. The type of nutrients ingested is unknown.

Answer: Thank you for highlighting this aspect. The study investigates the potential additional effect of a physical exercise program in liver transplant patients following nutritional programs based on the Mediterranean diet. All participants had a Medlite score > 8.5 (11.7 and 10.4), so good eating habits can be considered. In this context, we preferred to use a validated questionnaire to obtain a scientifically reliable score. Furthermore, we reasonably believe that the type of nutrients ingested does not add additional helpful information in line with the purpose of the study.

 Regarding the exercise method, it is also necessary to explain why this method was chosen.

Answer: Thanks for the comment. An additional sentence has been added in the 2.2 section between lines 97-100:

“An unsupervised physical exercise program was chosen to prevent participants from traveling to a specific health club to carry out the training to achieve a lower probability of absence from exercise and greater adherence to the program in the long run term.”

In addition, the IPAQ was performed to verify compliance with the exercise program.

  1. As a result, the relationship between exercise and diet is already known, and it is difficult to differentiate this from previous content.

Answer: Thanks for the comment. We agree with the reviewer about the relationship between diet and exercise. The study also aims to verify the additional effect of unsupervised physical exercise in subjects who all follow the Mediterranean diet in a population at high cardiovascular risk, such as transplant patients. The authors have added physical exercise to only half of the study sample to verify a specific differentiation between diet and physical exercise.

Furthermore, highly reliable tools, such as CPET and BIA, were used to verify any differences to evaluate cardiovascular performance and body composition.

  1. I don't understand whether you are referring to the heart or the liver.

Answer: Thanks for the comment. The study population consists of patients undergoing liver transplantation. As reported in the literature, this population is at high cardiovascular risk despite surgical treatment.

One of the main aspects after transplantation is the maintenance of a low cardiovascular risk profile induced by a sedentary lifestyle, pharmacological treatment, and any comorbidities.

Regular physical activity and diet are the non-pharmacological treatments frequently used in these categories of subjects.

For this reason, the multidisciplinary approach evaluates body composition and cardiovascular performance using CPET and nutritional habits. This approach was performed to preserve the integrity of the transplanted organ.

Reviewer 2 Report

Comments and Suggestions for Authors

Manuscript ID: nutrients-2769040

Article Report

Comments and Suggestions for Authors

The article wants to determine the intervention’s effectiveness of unsupervised physical

activity programs and the Mediterranean diet in male liver transplant recipients

compared to those who only follow Mediterranean diet recommendations. This study is

well-designed, 33 participants were divided into two groups, specifically 16 patients who

performed the unsupervised exercise program and 17 who adhered to nutritional

recommendations to the Mediterranean diet with a score <8.5. After the authors

evaluated the cardiopulmonary exercise testing (CPET) and body composition, they

observed the frailty in post-transplanted subjects and the importance of physical activity

in improving quality of life and reducing cardiovascular risk.

I. Minor comments:

1. Check double spaces between words, for example, lines 26, 29 (add a full stop

in the final sentence), 147, 336…

2. “Data are expressed as mean +- s.d.” is better to put it in the table foot rather

than in the title.

3. Line 104: add the meaning of “METs”.

4. Line 337: correct double “in”.

Comments on the Quality of English Language

Manuscript ID: nutrients-2769040

Article Report

Comments and Suggestions for Authors

The article wants to determine the intervention’s effectiveness of unsupervised physical

activity programs and the Mediterranean diet in male liver transplant recipients

compared to those who only follow Mediterranean diet recommendations. This study is

well-designed, 33 participants were divided into two groups, specifically 16 patients who

performed the unsupervised exercise program and 17 who adhered to nutritional

recommendations to the Mediterranean diet with a score <8.5. After the authors

evaluated the cardiopulmonary exercise testing (CPET) and body composition, they

observed the frailty in post-transplanted subjects and the importance of physical activity

in improving quality of life and reducing cardiovascular risk.

I. Minor comments:

1. Check double spaces between words, for example, lines 26, 29 (add a full stop

in the final sentence), 147, 336…

2. “Data are expressed as mean +- s.d.” is better to put it in the table foot rather

than in the title.

3. Line 104: add the meaning of “METs”.

4. Line 337: correct double “in”.

Author Response

Reviewer 2:

The authors would like to thank the reviewer for appreciating our work and suggestions provided to improve our manuscript.

Changes made to the manuscript are highlighted in red.

Below are the answers to the reviewer’s comments.

Minor

Comments and Suggestions for Authors

The article wants to determine the intervention’s effectiveness of unsupervised physical activity programs and the Mediterranean diet in male liver transplant recipients compared to those who only follow Mediterranean diet recommendations. This study is well-designed, 33 participants were divided into two groups, specifically 16 patients who performed the unsupervised exercise program and 17 who adhered to nutritional recommendations to the Mediterranean diet with a score <8.5. After the authors evaluated the cardiopulmonary exercise testing (CPET) and body composition, they observed the frailty in post-transplanted subjects and the importance of physical activity in improving quality of life and reducing cardiovascular risk.

  1. Minor comments:
  2. Check double spaces between words, for example, lines 26, 29 (add a full stop in the final sentence), 147, 336…

Answer: Thanks for the comment. Modifications were done accordingly.

  1. “Data are expressed as mean +- s.d.” is better to put it in the table foot rather than in the title.

Answer: Thanks for the comment. Modifications were done accordingly.

  1. Line 104: add the meaning of “METs”.

Answer: Thanks for the comment. Modifications were done accordingly.

  1. Line 337: correct double “in”.

Answer: Thanks for the comment. Modifications were done accordingly.